# Deep-Learning-Based Resource Allocation for Time-Sensitive Device-to-Device Networks

**DOI:** 10.3390/s22041551

**Published:** 2022-02-17

**Authors:** Zhe Zheng, Yingying Chi, Guangyao Ding, Guanding Yu

**Affiliations:** 1State Grid Key Laboratory of Power Industrial Chip Design and Analysis Technology, Beijing Smart-Chip Microelectronics Technology Co., Ltd., Beijing 102299, China; zhengzhe@sgitg.sgcc.com.cn (Z.Z.); chiyingying@sgitg.sgcc.com.cn (Y.C.); 2College of Information Science and Electronic Engineering, Zhejiang University, Hangzhou 310058, China; guangyaoding@zju.edu.cn

**Keywords:** 5G, ultra-reliable low-latency communications (URLLC), device-to-device (D2D) communications, resource allocation

## Abstract

Ultra-reliable and low-latency communication (URLLC) is considered as one of the major use cases in 5G networks to support the emerging mission-critical applications. One of the possible tools to achieve URLLC is the device-to-device (D2D) network. Due to the physical proximity of communicating devices, D2D networks can significantly improve the latency and reliability performance of wireless communication. However, the resource management of D2D networks is usually a non-convex combinatorial problem that is difficult to solve. Traditional methods usually optimize the resource allocation in an iterative way, which leads to high computational complexity. In this paper, we investigate the resource allocation problem in the time-sensitive D2D network where the latency and reliability performance is modeled by the achievable rate in the short blocklength regime. We first design a game theory-based algorithm as the baseline. Then, we propose a deep learning (DL)-based resource management framework using deep neural network (DNN). The simulation results show that the proposed DL-based method achieves almost the same performance as the baseline algorithm, while it is more time-efficient due to the end-to-end structure.

## 1. Introduction

The focus of the fifth-generation (5G) wireless communication networks is to provide reliable services for various applications with the design objectives of high throughput, reduced end-to-end latency, and massive device connectivity [1]. To support the emerging mission-critical applications (such as vehicle-to-vehicle communications, industrial automation, and virtual reality) that are sensitive to time delay and transmission reliability [2], it is necessary for wireless communication systems to achieve high quality of service (QoS). Therefore, ultra-reliable and low-latency communication (URLLC), which intrinsically takes reliability and latency into consideration during network design, is considered as one of the major use cases in 5G networks.

Aiming at achieving an ultra-low packet error rate (e.g., $10^{-5}$) and an ultra-low latency (e.g., 1 ms) [3,4], URLLC brings new challenges to the network design. Due to the inherent instability of wireless channel, it is usually difficult to achieve low latency and high reliability simultaneously. The reliability can be improved by allocating more radio resources, such as spectrum, re-transmissions, etc., which leads to the increase of transmission latency and the degradation of throughput performance [5]. Therefore, it is necessary to develop novel system architectures and network layouts to satisfy URLLC requirements by emphasizing high reliability and low latency simultaneously.

To support the increasingly heavy local traffic in wireless services, a device-to-device (D2D) communication enabled cellular network has been widely studied and is regarded as one of the key components in 5G networks. Due to the physical proximity of communicating devices, D2D communications can significantly improve the spectrum and energy efficiency [6]. In addition, the direct communication link between D2D pairs, as the most important feature of D2D communications, can also reduce the end-to-end latency of data transmission, which makes D2D communications a powerful tool to achieve URLLC in 5G networks. In general, a D2D network consists of several D2D pairs that reuse the spectrum of cellular networks. There are mainly two reusing modes in the existing literature, namely underlay mode and overlay mode [7]. In the underlay mode, D2D pairs transmit data on the spectrum allocated to cellular users (CUs), resulting in the interference between D2D pairs and CUs. To improve the system performance of the underlay mode, some methods have been developed to alleviate the co-channel interference, including mode selection [8] and resource allocation [9]. In the overlay mode, a dedicated spectrum is allocated to D2D pairs to allow interference-free D2D transmissions. In this paper, we focus on the overlay mode to avoid the performance degradation of CUs. Although D2D communications can be conducted without interference in the overlay mode, the reserved spectrum is usually rather limited, which makes it infeasible to allocate spectrum orthogonally for D2D pairs. Multiple D2D pairs may transmit data on the same channel with severe co-channel interference, resulting in the degradation of system performance [10]. Therefore, it is necessary to determine the proper channel and power allocation for D2D pairs to guarantee the system performance, including throughput, latency, and reliability.

### 1.1. Related Works

The radio resource allocation problem has been widely studied to improve the system performance of D2D networks. In [11], mode selection and channel-power allocation have been both considered to improve the throughput performance of D2D networks. In [12], a matching algorithm based on game theory has been proposed to improve the energy efficiency of D2D transmission. In [13], the sum-rate is maximized under different rate requirements of D2D pairs. Although several system metrics including throughput and energy efficiency have been widely investigated in the existing works, there are not sufficient studies on the URLLC performance of D2D networks. In [14], a power control policy has been proposed to achieve the maximum average rate in the finite block-length regime with a reliability constraint. However, the system model in this work is too simple, where the system only consists of a D2D pair and a cellular user. Leveraging the game theory, the successful transmission probability of an overlay D2D network is maximized by achieving Nash equilibrium [15]. Nevertheless, the power control strategy is not investigated, and the throughput performance of the D2D network is not guaranteed. In [16], the authors have proposed a joint reservation and contention-based channel allocation strategy to minimize the sum packet error probability of D2D networks. In [17], the authors have proposed a D2D-based protocol to improve the reliability of downlink transmissions. In [18], a two-timescale resource allocation framework has been proposed, where the user pairing is conducted at the long timescale and the transmission time is determined at the short timescale.

In recent years, deep learning (DL) has been successfully applied to a variety of fields, such as computer vision, natural language processing, and speech recognition. In the communication systems, a variety of DL-based methods have also been used in resource allocation [19], user association [20], channel estimation [21], and physical layer processing [22]. In [23], a resource allocation strategy based on deep neural network (DNN) has been designed to increase the overall spectrum efficiency of an underlay D2D network. In [24], the channel selection and transmit power of the D2D pairs are jointly optimized to achieve a higher weighted sum rate by using deep reinforcement learning. However, the DL-based resource management method for URLLC-oriented D2D networks has not been investigated in the existing literature.

### 1.2. Motivation and Contribution

Although the existing methods can improve the URLLC performance of D2D networks by properly allocating the radio resources, the throughput performance of D2D networks at the finite blocklength has not been investigated. Moreover, the high computational complexity brought by the iterative structure may lead to the degradation of latency performance, which motivates us to develop a time-efficient resource management approach for D2D networks based on DL. To this end, in this paper, we propose a resource allocation method for overlay D2D networks based on DL. The main contributions of this paper are summarized as follows.

Instead of Shannon rate, we adopt the short packet coding rate to more accurately capture the rate loss in the finite blocklength regime. The reliability of D2D transmissions is guaranteed by choosing a proper coding rate lower than the achievable rate.We propose an iterative channel selection and power allocation algorithm based on game theory. The sum utility function of all D2D pairs is maximized by alternately updating the channel selection and power allocation of each D2D pair.To improve the time efficiency of the resource allocation procedure, we propose two DNN-based methods to solve the sum rate maximization problems with and without the minimum rate constraint, respectively. The network structure and output are properly designed to improve the learning ability of the network.

The rest of this paper is organized as follows. Section 2 introduces the system model and formulates the sum rate maximization problem for D2D networks. In Section 3, the game theory-based algorithm is proposed. In Section 4, two DNN-based methods are proposed for resource allocation in D2D networks. In Section 5, simulation results are provided to demonstrate the effectiveness of the proposed algorithms. Finally, the article is concluded in Section 6.

## 2. System Model and Problem Formulation

### 2.1. System Model and Transmission Model

In this paper, we consider an overlay multi-channel D2D network, as depicted in Figure 1. In particular, *N* D2D pairs are arbitrarily distributed and share *K* orthogonal wireless channels, where *N* is typically larger than *K*. Each D2D pair consists of one transmitter and one receiver. Let N={1,⋯,N} denote the set of D2D pairs and K={1,⋯,K} denote the set of channels. We consider a slotted OFDM transmission scheme where the length of a time slot is *T* and the bandwidth of a channel is *B*. Due to the strict latency requirement of the D2D pairs, *T* is assumed to be rather small to shorten the transmission latency.

Let $h_{n,m}^{k}=\alpha_{n,m}g_{n,m}^{k}$ denote the channel gain of the *k*-th channel between the transmitter of D2D pair *n* and the receiver of D2D pair *m*, where $\alpha_{n,m}$ and $g_{n,m}^{k}$ are the large-scale and small-scale channel fading, respectively. The large-scale channel fading is composed of path loss and shadowing, as $\alpha_{n,m}=G\psi_{n,m}d_{n,m}^{-\phi }$, where *G* is the pathloss constant, $\psi_{n,m}$ is the log-normal shadowing random component, $d_{n,m}$ is the distance between the transmitter of D2D pair *n* and the receiver of D2D pair *m*, and ϕ represents the path loss exponent. The small-scale channel gain $g_{n,m}^{k}$ is assumed to be Rayleigh fading, which is independent and identically distributed (i.i.d) with unit mean for different n,m, and *k*. It is noted that $h_{n,m}^{k}$ represents the local D2D channel when n=m and represents the interference D2D channel when n≠m. In practical communication systems, the real-time channel state information (CSI) is hard to obtain due to the dynamic wireless environment. To tackle this problem, the accurate CSI could be substituted with the position information of the D2D pairs, which is much easier to obtain. We assume that the position of the D2D pairs does not change in a certain time interval. Therefore, the average large-scale channel gain can be calculated as $Gd^{-\phi }$. By using the large-scale channel gain as the input of the proposed methods, the resource allocation strategy could be obtained in the absence of accurate CSI.

It is assumed that each D2D pair can only occupy a single resource block (RB) for data transmission, which can relieve the channel congestion and reduce the computational complexity at the receiver. Let $\rho_{n}^{k}$ be the channel indicator of D2D pair *n*; i.e., $\rho_{n}^{k}=1$ indicates that D2D pair *n* is transmitting data through channel *k* and $\rho_{n}^{k}=0$ otherwise. Then, the SINR of D2D pair *n* can be expressed as
(1)$$\gamma_{n}=\frac{\sum_{k=1}^{K}\rho_{n}^{k}p_{n}h_{n,n}^{k}}{\sum_{k=1}^{K}\sum_{l=1,l\neq n}^{N}\rho_{n}^{k}\rho_{l}^{k}p_{l}^{k}h_{l,n}^{k}+N_{0}B},$$
where $p_{n}$ is the transmit power of the D2D pair *n* and $N_{0}$ is the power spectral density of noise.

Due to the strict latency constraint of URLLC services, the slot duration of D2D transmission is very short. Therefore, the traditional Shannon channel capacity that assumes the code length to be infinite does not hold in this case. With the bandwidth *B* and slot duration *T*, the maximal achievable rate of D2D pair *n* in the finite blocklength regime is accurately approximated by [25]
(2)$$R_{n}=B\log_{2}\left(1+\gamma_{n}\right)-\sqrt{\frac{BV_{n}}{T}}\frac{Q^{-1}\left(\epsilon \right)}{\ln2},$$
where $Q^{-1}\left(\cdot \right)$ is the inverse of the Gaussian Q-function, ϵ is the packet error probability, and $V_{n}$ is the channel dispersion given by [25]
(3)$$V_{n}=1-{\left(1+\gamma_{n}\right)}^{-2}\approx 1.$$

According to [26], the approximation in (Equation 3) is quite accurate in the high SNR regime.

### 2.2. Problem Formulation

In the D2D network, each D2D pair chooses a single channel and a certain transmit power according to the global channel information. We first formulate the achievable rate maximization problem without minimum rate constraint as
(4a)$$\begin{array}{cc} P1:\;\max_{\left\{\rho_{n}^{k},P_{n}\right\}}\; & \sum_{n=1}^{N}R_{n}, \end{array}$$
(4b)$$\begin{array}{cc} s.t.\;\; & \sum_{k=1}^{K}\rho_{n}^{k}=1,\;\forall n\in N,\forall k\in K, \end{array}$$
(4c)$$\begin{array}{cc} & \rho_{n}^{k}\in \left\{0,1\right\},\;\forall n\in N,\forall k\in K, \end{array}$$
(4d)$$\begin{array}{cc} & p_{n}⩽P_{\max},\;\forall n\in N, \end{array}$$
(4e)$$\begin{array}{cc} & p_{n}⩾0,\;\forall n\in N. \end{array}$$

The objective of problem P1 is to maximize the sum finite blocklength rate of all D2D pairs, where constraints (4b), (4c) ensure that each D2D pair can only choose a single channel and (4d), (4e) are the transmit power constraints.

In some cases, the D2D pairs may have a minimum rate requirement on the communication links to achieve better QoS. To address this requirement, the achievable rate maximization problem with a minimum rate constraint can be formulated as
(5a)$$\begin{array}{cc} P2:\;\max_{\left\{\rho_{n}^{k},P_{n}\right\}}\; & \sum_{n=1}^{N}R_{n}, \end{array}$$
(5b)$$\begin{array}{cc} s.t.\; & R_{n}⩾R_{n}^{\min},\;\forall n\in N,\\ & (4b)-(4e). \end{array}$$

Problems P1 and P2 are both mixed integer nonlinear programming (MINLP) problems with non-convex objective functions, which are hard to solve. Although some heuristic algorithms can be applied to find the suboptimal solution, their iterative structures with high computational complexity may lead to the degradation of latency performance. The classic branch-and-bound (BnB) algorithm can find global optimal solutions, but its complexity scales exponentially with the problem size. In this problem, the complexity of the BnB algorithm grows significantly as the user number increases, which is challenging for real-time implementation. On the other hand, the recent successful applications of machine learning techniques demonstrate that DL has the ability to learn various patterns and approximate arbitrary function mappings [27]. Moreover, the time complexity of the DL method is usually much lower than that of the classic methods, which makes DL a competitive tool when solving communication problems. In fact, DL has been successfully applied to the channel selection and power control in D2D networks [23,24]. Therefore, we adopt the powerful DL method to solve the formulated MINLP problem by directly mapping the input data to the channel selection and power allocation strategy.

## 3. Game Theory-Based Resource Allocation

Since the complexity of the global optimal algorithm scales exponentially with the problem size, it is necessary to propose a suboptimal solution with low computational complexity. In this section, we propose a channel selection and power allocation algorithm based on spatial adaptive play (SAP), which is a high-profile algorithm in potential games because it can achieve the optimal Nash equilibrium. In fact, SAP has been successfully applied to the resource allocation problem in wireless communications [15], which motivates us to utilize SAP to solve the formulated problems. In SAP, one player is randomly chosen to perform learning in each iteration. The sum utility function of all D2D pairs is maximized by alternately updating the channel selection and power allocation of each D2D pair. To reduce the computational complexity, the transmit power is discretized into *L* discrete levels, that is, $p_{n}\in P=\left\{P_{1},P_{2},\cdots P_{L}\right\}$. For simplicity, let $P=\left\{p_{1},p_{2},\cdots ,p_{N}\right\}$ denote the power vector that collects the transmit power of all D2D pairs. Let Φ denote the N×K-sized channel selection matrix where $\Phi_{n,k}=\rho_{n}^{k}$. Let *H* denote the N×N×K-sized channel matrix where $H_{n,m,k}=h_{n,m}^{k}$.

The proposed algorithm for problems P1 and P2 is shown in Algorithm 1. In step 2, μ is the exploration probability, which represents the willingness of exploration when the D2D pairs choose the channel and power level in step 6. When μ decreases, the D2D pairs are less willing to explore and tend to choose the channel and power level with the highest utility function. *t* is the iteration index, *w* is the decay rate that controls the decreasing speed of μ, and τ is the aborting threshold. $U^{t}$ is the value of the utility function in the *t*-th iteration, which is different for problems P1 and P2. To be specific, for problem P1, $U=\sum_{n=1}^{N}R_{n}$ is the sum rate of D2D pairs. For problem P2, $U=\sum_{n=1}^{N}\left[R_{n}-\eta \max\left(R_{n}^{min}-R_{n},0\right)\right]$ is the penalized sum rate of D2D pairs, where η>0 is a weight constant to ensure that the minimum rate constraint is satisfied. As η increases, the minimum rate constraint is better guaranteed, while the sum rate decreases because the interference becomes more severe. Therefore, the rate requirement of the users with bad channel condition could be satisfied by adopting a large η, although the sum rate performance might be sacrificed in this case.
**Algorithm 1** SAP-Based Channel Selection and Power Allocation Algorithm1:**Input:**$H,N,K,B,T,\epsilon ,\sigma^{2}$.2:**Initialization:** Set t=0, μ=1, w<1, τ>0, and $U^{0}=0$. Initialize the channel selection matrix Φ and power matrix *P*.3:**Repeat:**4:• Set t=t+1.5:• Randomly choose a D2D pair *n*, calculate the sum rate on all possible channel selection $\rho_{n,k},k\in K$ and power allocation $p_{n}$. Assume that the utility function is maximized as $U^{t}$ when D2D pair *n* chooses channel $k_{0}$ with power level $l_{0}$.6:• With probability 1−μ: Set $\rho_{n}^{k_{0}}=1$, $\rho_{n}^{k}=0,k\neq k_{0}$, and $p_{n}=P_{l_{0}}$. With probability μ: Set $\rho_{n}^{k_{1}}=1$, $\rho_{n}^{k}=0,k\neq k_{1}$, and $p_{n}=P_{l_{1}}$, where $k_{1}$ and $l_{1}$ are randomly picked from $\left\{1,\cdots ,K\right\}$ and $\left\{1,\cdots ,L\right\}$, respectively.7:• Set μ=wμ.8:**Until**$|U^{t}-U^{t-1}|<\tau$.9:**Output:**Φ and *P*.

In each iteration, the algorithm randomly chooses a D2D pair and finds the optimal channel selection and power allocation strategy through exhaustive search. In this way, the D2D pairs take turns to update the strategy to increase the sum rate step by step, which guarantees the convergence of the algorithm. On the other hand, the D2D pairs explore the possible actions with probability μ in each iteration. This can help the algorithm jump out of the local minimum and find better solutions. It is noted that the exploration probability keeps decreasing as the number of iterations increases, which means that the algorithm gradually stops exploring and finally converges to the solution.

After a number of iterations, the channel selection and power allocation of all D2D pairs will converge to a Nash equilibrium, which means that no D2D pair can achieve a higher value of the utility function by changing its channel selection and transmit power. The channel selection matrix Φ and power matrix *P* at this point are taken as the final output of the algorithm. It is noted that it takes a number of iterations for the SAP-based channel selection and power allocation algorithm to converge. The non-ignorable computational overhead may lead to the degradation of latency performance. Therefore, it is necessary to develop a time-efficient DL-based method to solve the channel selection and power allocation problem.

## 4. Deep Learning Based Resource Allocation

In this section, we develop a DNN-based channel selection and power allocation method for problems P1 and P2. In the proposed DNN model, a basic building block is composed of a fully connected (FC) layer, a batch normalization (BN) layer, and a rectified linear unit (ReLU) layer. The structures of the DNN models and the training process will be discussed in the following subsections.

### 4.1. Basic DNN Module

In the DNN model, a basic building block is composed of a fully connected (FC) layer, a batch normalization (BN) layer, and a rectified linear unit (ReLU) layer.

In the FC layer, the input vector is multiplied with a weight matrix, and the result is added with a bias vector to form the output vector. Let *x* and $y_{\mathrm{FC}}$ denote the $N_{\mathrm{in}}$-sized input vector and the $N_{\mathrm{out}}$-sized output vector of the FC layer, respectively. Then, the relationship between *x* and $y_{\mathrm{FC}}$ is given by $y_{\mathrm{FC}}=Wx+b$, where *W* is the $N_{\mathrm{out}}\times N_{\mathrm{in}}$-sized weight matrix and *b* is the $N_{\mathrm{out}}$-sized bias vector.

Then, $y_{\mathrm{FC}}$ passes the BN layer, which normalizes the data by re-centering and re-scaling in a mini-batch manner. To be specific, during the training process, multiple samples are fed into the DNN simultaneously as a mini-batch to improve the training efficiency and accelerate the converging. The function of the BN layer is normalizing the input data by its mean and variance. Let $y_{\mathrm{BN}}$ denote the $N_{\mathrm{out}}$-sized output vector of the BN layer. Then, $y_{\mathrm{BN}}=\gamma \frac{y_{\mathrm{out}}-\mu_{B}}{\sqrt{\sigma_{B}^{2}+\epsilon_{0}}}+\beta$, where $\mu_{B}$ and $\sigma_{B}^{2}$ are the mean and variance of the input mini-batch, $\epsilon_{0}$ is an arbitrarily small constant for numerical stability, and γ and β are learnable parameters of the BN layer. By normalizing the distribution of the input data, the BN layer can help solve the vanishing gradient problem and avoid overfitting.

Finally, $y_{\mathrm{BN}}$ enters the ReLU layer, which performs nonlinear transformation to enhance the learning capacity of DNN, where $y_{\mathrm{RelU}}=\max\left(y_{\mathrm{BN}},0\right)$.

### 4.2. DNN Model for Resource Allocation

In this subsection, we propose a DNN based method to solve the sum rate maximization problem P1. The network structure is shown in Figure 2. The N×N×K-sized channel matrix is flattened into a channel vector of length $N^{2}K$ in the reshaping stage. In the normalization stage, the channel vector is first converted to the dB scale and then normalized to have zero mean and unit variance.

After reshaping and normalization, the channel information enters the feature network, which is composed of $M_{F}$ DNN blocks, each of which consists of $Q_{F}$ neurons. The function of the feature network is encoding the channel information into deep features, which can be leveraged by the channel network and power network.

The channel network is composed of $M_{C}$ DNN blocks and a softmax layer, which takes the output of the feature network as the input. Each DNN block consists of $Q_{C}$ neurons. The output vector of the last DNN block is NK-sized, which is divided into *NK*-sized vectors and fed into the partial softmax layer. Let $y_{n}^{k}$ denote the *k*-th element in the *n*-th group. After passing the softmax layer, the probability value corresponding to $y_{n}^{k}$ is given by
(6)$${\hat{y}}_{n}^{k}=\frac{e^{y_{n}^{k}}}{\sum_{k=1}^{K}e^{y_{n}^{k}}},$$
which denotes the probability that D2D pair *n* chooses channel *k* for data transmission. In the evaluating phase, each D2D pair will choose the channel with the largest probability value, the index of which is converted into a one-hot vector representing the channel selection result.

Composed of $M_{P}$ DNN blocks and a softmax layer, the power network has a structure similar to the channel network. Each DNN block consists of $Q_{P}$ neurons. The output vector of the last DNN block is NL-sized, which is divided into *NL*-sized vectors and fed back into the partial softmax layer. Let $p_{n}^{l}$ denote the *l*-th element in the *n*-th group. After passing the softmax layer, the probability value corresponding to $p_{n}^{l}$ is given by
(7)$${\hat{p}}_{n}^{l}=\frac{e^{p_{n}^{l}}}{\sum_{l=1}^{L}e^{p_{n}^{l}}},$$
which denotes the probability that D2D pair *n* chooses power level *l* for data transmission. In the evaluating phase, each D2D pair will choose the power level with the largest probability.

The whole network is trained in an unsupervised manner, where the inverse sum rate of all D2D pairs is taken as the loss function
(8)$$L1=-\sum_{n=1}^{N}R_{n}.$$

During the training phase, the channel selection and power allocation based on the network output is different from that of the evaluating phase. In fact, the discretized channel selection and power allocation result cannot be applied to the loss function, because the gradient cannot be calculated during the backpropagation. To address this problem, instead of choosing the channel with the largest ${\hat{y}}_{n}^{k}$, D2D pair *n* directly chooses the channel as $\rho_{n}^{k}={\hat{y}}_{n}^{k}$. Here, $\rho_{n}^{k}$ represents the proportion of power that D2D pair *n* allocates to the *k*-th channel. On the other hand, the transmit power of D2D pair *n* is given by $p_{n}=\sum_{l=1}^{L}{\hat{p}}_{n}^{k}P_{l}$, where ${\hat{p}}_{n}^{l}$ is regarded as the weight of the *l*-th power level. Finally, we discuss the computational complexity of the DNN model by calculating the number of multiplications during the forward propagation. In fact, the number of multiplications of a basic DNN module is $N_{\mathrm{out}}\times N_{\mathrm{in}}$. Therefore, the overall number of multiplications is $M_{F}Q_{F}^{2}+M_{C}Q_{C}^{2}+M_{P}Q_{P}^{2}$.

### 4.3. DNN Model for Resource Allocation with Minimum Rate Constraint

In this subsection, we propose a DNN-based method to solve the sum rate maximization problem P2, which has a minimum rate constraint. The network structure is shown in Figure 3. An integrated network is composed of a feature network, a channel network, and a power network. In network 1, the pre-processed channel information is taken as the input, while the channel information together with the output of the previous network is taken as the input in the following networks. The structure of network 2-*M* is the same as that of network 1, which is omitted in the figure for simplicity. The only difference is that the input data of network 2-*M* is composed of the channel gain vector, the output of the previous channel network, and the output of the previous power network.

The network is trained in an unsupervised manner, where the inverse sum rate of all D2D pairs with minimum rate penalty is taken as the loss function
(9)$$L2=-\sum_{n=1}^{N}\left[R_{n}-\eta \max\left(R_{n}^{min}-R_{n},0\right)\right],$$
where η>0 is a weight constant to realize the tradeoff between the sum rate and the minimum rate constraint. Similar to the SAP-based method, the minimum rate constraint is better guaranteed as η increases, while the sum rate decreases because the interference becomes more severe. The channel selection and power allocation based on the network output are the same as the DNN model for problem P1 in the previous subsection. During the training phase, the whole network is trained step by step. To be specific, network 1 is first trained by calculating loss L2 using the output of network 1. After that, network 1 and network 2 are trained together by calculating loss L2 using the output of network 2. The following networks are trained in this manner as well. In this way, the loop-like structure of the network can better deal with the minimum rate constraint of the D2D pairs. To satisfy the minimum rate constraint more strictly, the channel selection and power allocation result of the network is further adjusted by the SAP-based method with a small number of iterations. The computational complexity is approximately *M* times the complexity of the DNN model for problem P1. To be specific, the overall number of multiplications is approximately $M(M_{F}Q_{F}^{2}+M_{C}Q_{C}^{2}+M_{P}Q_{P}^{2})$.

## 5. Simulation Results

In this section, simulations are conducted to validate the performance of the proposed DNN-based resource allocation algorithm for D2D networks. The D2D pairs are randomly distributed in an area of 100 m × 100 m. The maximum distance between the transmitter and the receiver of each D2D pair is set to 15 m. The path-loss model is adopted as $128.1+37.6\log\left(d\;\left[\mathrm{km}\right]\right)$, the standard deviation of the log-normal shadowing distribution is 8 dB, and the Rayleigh fading is also adopted. Before training the network, 10,000 channel samples are generated, which are divided in a train set and a test set with the ratio of 4:1. The learning rate during training is set to be 0.001 and the Adam optimizer is adopted. Other simulation parameters are listed in Table 1. The URLLC-related parameters are the slot length *T*, the reliability constraint ϵ, and the minimum rate $R_{\min}$. These parameters are set according to [3,4], where small packets in the order of 100 bits must be delivered within no more than 1 ms and the packet error probability is no larger than $10^{-5}$. Other parameters follow the simulation settings of the previous works [23,24].

The convergence curve of the proposed DNN-based methods is shown in Figure 4, where the number of channels is set to 5. The performance of the SAP-based method is tested as the baseline. It can be observed that the sum rates of the DNN-based methods manage to converge after training for around 10 epochs. For the sum rate maximization problem P1, the highest sum rate of the DNN-based method is slightly lower than that of the SAP-based method. For problem P2 with a minimum data rate constraint, the sum rates of both methods are much lower than those of problem P1. This is because the minimum data rate constraint leads to more active D2D pairs, and the sum rate performance degrades due to higher interference. It is noted that the sum rate of the DNN-based method for problem P2 first increases steadily, then decreases gradually, and finally approaches that of the SAP-based method. The reason is that the minimum rate constraint is more difficult to learn than the sum rate performance. Therefore, the DNN model first learns to improve the sum rate performance and then learns to satisfy the minimum rate constraint gradually. On the other hand, the sum rate of the SAP-based method is lower than that of the DNN-based method, because the minimum rate constraint is more strictly satisfied in the SAP-based method due to its iterative structure, which leads to higher interference among D2D pairs.

Figure 5 demonstrates the sum rate performance of the proposed methods for problem P1 with different numbers of channels. As the channel number *K* increases, the sum rates of both methods increase. We can also observe that the SAP-based method provides a higher sum rate than the DNN-based method. Figure 6 depicts the complementary cumulative distribution functions (CCDFs) of the transmission rate for problem P2. Here, the minimum rate $R_{\min}$ is set to 0.1 Mbps, and the channel number *K* is 5. From the figure, about 80% of D2D pairs can achieve a transmission rate larger than $R_{\min}$ in the SAP-based method. In comparison, the proportion in the DNN-based method is about 75%. As the transmission rate increases, the gap between two methods becomes smaller. From Figure 5 and Figure 6, the SAP-based method achieves better performance than the DNN-based method. However, the gap between the performance of two methods is very small, and the DNN-based method is more time-efficient due to the iteration-free structure.

Table 2 compares the SAP-based method and the DNN-based method with regard to the sum rate performance and the rate excess probability. To be specific, the rate excess probability is the probability that the rate of a D2D pair is larger than $R_{\min}$. From the table, it is observed that the sum rate of the SAP-based method is higher than that of the DNN-based method for Problem P1. In comparison, the sum rate of the SAP-based method is lower than that of the DNN-based method for Problem P2. This is because the SAP-based method can better guarantee the minimum rate requirement due to its iterative structure, which results in the rate loss compared with the DNN-based method.

## 6. Conclusions

This paper develops a novel resource allocation algorithm for overlay D2D networks based on DNN to support URLLC services. We first formulate two sum rate maximization problems with and without minimum rate constraint. Then, we design an iterative algorithm based on game theory as the baseline. After that, a DNN-based channel selection and power allocation method is developed. The network is trained in an unsupervised manner with no need for labeled data, and two different loss functions are designed for the two formulated problems, respectively. Finally, the performance of the proposed methods is demonstrated through numerical simulation.

In fact, the proposed method could be adapted from different aspects so that it could be applied in a practical system. For example, transfer learning could be applied to accelerate the training process under different scenarios. The detailed implementation of the proposed methods in a real D2D 5G environment will be investigated in our future work.

## Figures and Tables

**Figure 1 sensors-22-01551-f001:**
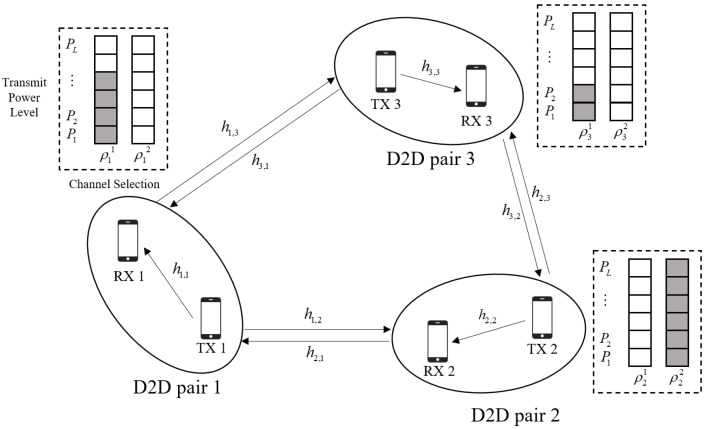
The system model of D2D networks.

**Figure 2 sensors-22-01551-f002:**
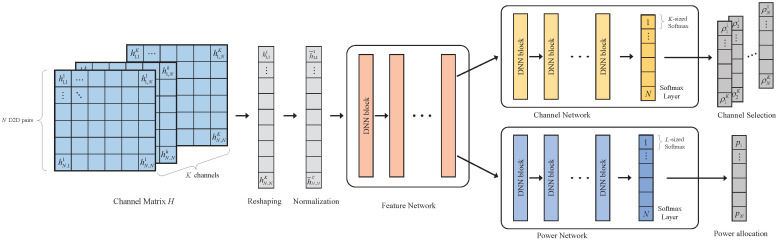
The DNN model for problem P1.

**Figure 3 sensors-22-01551-f003:**
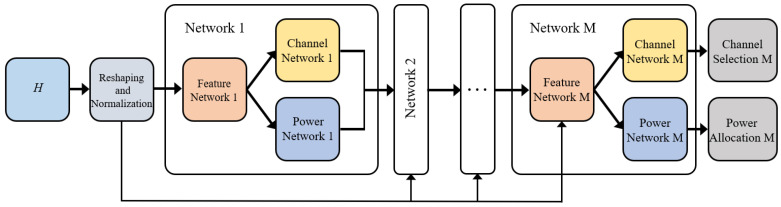
The DNN model for problem P2.

**Figure 4 sensors-22-01551-f004:**
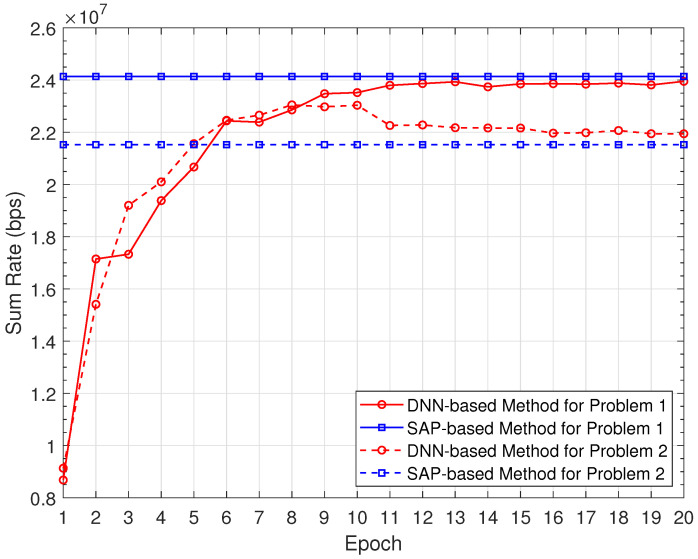
Convergence curve of the DNN-based methods.

**Figure 5 sensors-22-01551-f005:**
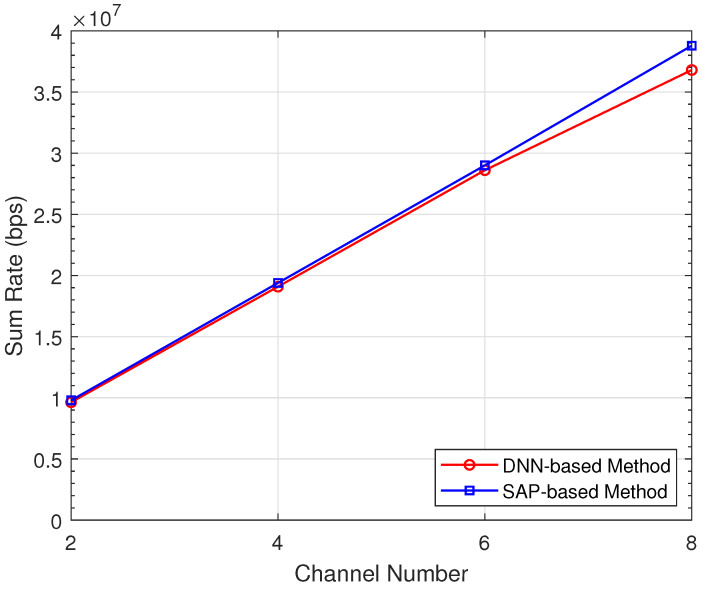
Sum rate performance for problem P1 with varying channel number.

**Figure 6 sensors-22-01551-f006:**
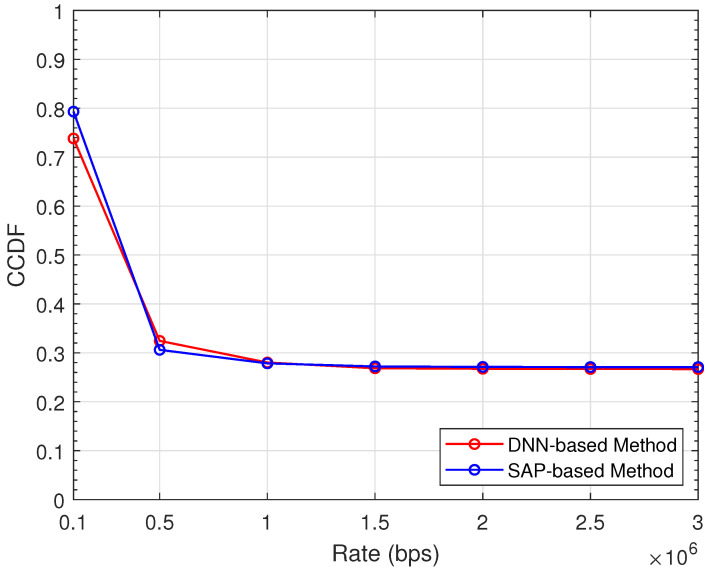
CCDF of the transmission rate for problem P2.

**Table 1 sensors-22-01551-t001:** Simulation parameters.

Parameter	Value
Number of D2D pairs, *N*	10
Number of channels, *K*	2,4,5,6,7
Channel bandwidth, *B*	180 KHz
Slot length, *T*	1 ms
Maximum transmit power, $P_{\max}$	23 dBm
Power spectral density of noise, $N_{0}$	−174 dBm/Hz
Reliability constraint, ϵ	$10^{-5}$
Minimum rate, $R_{\min}$	$10^{5}$ bps
Initial exploration probability μ	0.5
Decay rate *w*	0.9

**Table 2 sensors-22-01551-t002:** Performance comparison of the SAP-based method and the DNN-based method.

Algorithm	Sum Rate of P1	Sum Rate of P2	Rate Excess Probability of P2
DNN-based Method	23.84 Mbps	21.90 Mbps	0.7382
SAP-based Method	24.32 Mbps	21.52 Mbps	0.7903

## Data Availability

Data sharing not applicable.

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
