# Peer review of "Deep-Learning-Based Resource Allocation for Time-Sensitive Device-to-Device Networks"

_sensors, 2022, doi:10.3390/s22041551_

Round 1
Reviewer 1 Report
In this paper, the authors proposed game theoretic and DNN based approaches for overlay D2D communications for URLLC applications. Overall, the paper is well written, however it needs some corrections as follows:
- Algorithm 1 needs more explanations to show how it works as well as the motivation behind it.
- The resolution of figure 2 needs to be enhanced.
- Study of sum rate against power allocation need to be included.
- The paper needs to be extensively revised against typos and grammatical errors, to name few:
deep deep neural network (DNN) ------> deep neural network (DNN)
simulation results shows -------------> simulation results show
which aims at minimizing -----------> which aim at minimizing
strategy has been proposed ---------> strategy have been proposed
Author Response
We would like to express our sincere gratitude to the Editor and all the reviewers for providing insightful comments and constructive suggestions. These comments and suggestions helped us to significantly improve the quality of the manuscript. In the revised manuscript, we address the comments identified by the Editor and Reviewers with blue font. The references cited in the response refer to the references in the manuscript. Our detailed point-to-point responses to the comments of the editor and each reviewer are given in the following. We hope our response will satisfactorily address the reviewers' comments.

Reviewer 2 Report
You state while analyzing Figs 5 and 6 that SAP-based method achieves better performance than the DNN-based method. It is not clearly visible form Figures. Can You provide more evidence that straightens this claim, maybe by providing additional Figure or Table.
Author Response

(The authors gave the same response as above.)

Reviewer 3 Report
This paper considers resource allocation for time-sensitive device-to-device networks. This paper formulates resource allocation problem and designs a game theory based algorithm. In addition, this paper proposes a deep learning based resource management framework using deep neural network. Simulation results show that the deep learning method achieves almost the same performance as the game theory algorithm. However, the contribution of this paper is rather weak and the descriptions are poor. The paper needs significant improvement.
Although this paper focuses on time-sensitive device-to-device networks and ultra-reliable and low-latency communication, the problem formulation considers neither latency nor outage probabilities. Latency requirement (for example, less than 1 micro second) and block error rate requirement (for example, less than 0.00001) should be included in the equations.
In time-sensitive device-to-device networks, there is no time to exchange channel state information between devices. Hence, the global optimization problems in P1 and P2, which require channel information, do not seem to be feasible.
The problem formulations P1 and P2 are very unrealistic, since they only maximizes throughput and does not consider fairness. Although P2 includes minimum rate constraint, still users with bad channel condition need to be sacrificed. Without considering fairness, the problem formulations are too simple and rather meaningless.
Game theory based algorithm can achieve Nash equilibrium but, in many cases, game theory based solutions are far from global optimal solution. If the authors believe that a global optimal solution to P1 and P2 can be achieved through the game theory algorithm, the authors need to prove that. I don't understand why the authors tried game-theory based algorithm as a baseline. There are many other heuristic algorithms which can produce satisfactory results.
Since game theory based algorithm usually produce not-so-good results, it doesn't make sense to compare the deep learning based algorithm against the game theory based algorithm. The authors need to compare with the optimal solution or with other good heuristic solutions.
The deep learning based algorithm requires channel matrix of device-to-device networks as inputs. However, such channel matrix cannot be obtained in time-sensitive device-to-device networks, since there is no time to exchange channel state information.
Author Response

(The authors gave the same response as above.)

Round 2
Reviewer 1 Report
The authors addressed all the suggested comments
Author Response
Many thanks for your precious time for the review of our paper and for providing us with valuable comments and suggestions.
Reviewer 3 Report
No further comments.
Author Response

(The authors gave the same response as above.)
